# Role of Thalamic Ca_V_3.1 T-Channels in Fear Conditioning

**DOI:** 10.3390/ijms26083543

**Published:** 2025-04-09

**Authors:** Tamara Timic Stamenic, Srdjan M. Joksimovic, Brier Fine-Raquet, Vasilije P. Tadic, Vesna Tesic, Vesna Jevtovic-Todorovic, Slobodan M. Todorovic

**Affiliations:** 1Department of Anesthesiology, Anschutz Medical Campus, University of Colorado, Aurora, CO 80045, USA; joksimovis@chop.edu (S.M.J.); bfine1606@gmail.com (B.F.-R.); vasatadic@gmail.com (V.P.T.); vesna.tesic@lsuhs.edu (V.T.); vesna.jevtovictodorovic@cuanschutz.edu (V.J.-T.); slobodan.todorovic@cuanschutz.edu (S.M.T.); 2Neuroscience and Pharmacology Graduate Program, Anschutz Medical Campus, University of Colorado, Aurora, CO 80045, USA

**Keywords:** central medial nucleus of thalamus, T-type calcium channels, fear conditioning

## Abstract

The potential contribution of the ion channels that control the excitability of the midline and intralaminar nuclei of the thalamus to the modulation of behaviors has not been well studied. In this study, we used both global genetic deletion (knock-out, KO) and thalamus-specific molecular knock-down (KD) approaches to investigate the role of thalamic Ca_V_3.1 T-type calcium channels (T-channels) in fear learning and fear responses. Previously, we have shown that the dominant subtype of T-channels in the central medial nucleus of the thalamus (CMT) is the Ca_V_3.1 isoform and that CMT neurons from Ca_V_3.1 KO animals have decreased burst firing. By specifically knocking down Ca_V_3.1 T-channels in the CMT using the shRNA approach, we also reduced burst firing without affecting the tonic firing mode of the transfected neurons. We report that global Ca_V_3.1 KO animals showed stronger freezing behaviors during both the conditioning and testing phases of contextual fear conditioning, while CMT-specific Ca_V_3.1 KD mice only had stronger fear responses during testing. In contrast, the cue-mediated fear responses were similar between Ca_V_3.1 KO and Ca_V_3.1 KD mice and the controls. Our findings validate thalamic Ca_V_3.1 T-channels as a potential new target for the development or treatment of different psychiatric diseases, such as post-traumatic stress disorder, schizophrenia, anxiety, and substance abuse disorders.

## 1. Introduction

Contextual fear conditioning (CFC) is a form of Pavlovian learning in which an association is made between non-threatening environmental stimuli and painful, dangerous, or threatening stimulation [1,2]. Studies of Pavlovian fear conditioning and extinction in rodents and humans have suggested that a neural circuit including the hippocampus, amygdala, and medial prefrontal cortex is involved in the learning and memory processes that enable context-dependent behavior [3]. Recently, the roles of other parts of the brain in fear learning, responses, and extinction have been more extensively studied [4,5,6,7]. Although an early study showed that thalamic lesions did not prevent the acquisition of conditioned auditory fears [8], later studies found that large lesions in the posterior thalamus (including posterior intralaminar nuclei) prevented the acquisition of fear-potentiated startle [9,10]. Recently, the thalamus (midline and intralaminar nuclei) and its role in conditioned fear behaviors have received more attention. A part of the midline thalamus, mediodorsal thalamic nucleus (MD), has been implicated in the control of fear memory processes [11,12]. Specifically, it has been shown that the tonic firing in MD neurons contributes to extinction learning, whereas increased burst firing has a suppressive effect on fear extinction [11,12]. The paraventricular thalamus (PV, part of the midline thalamus) has been established as having an important role in the consolidation and retrieval of fear memories [13,14,15,16]. The PV–amygdala pathway constitutes a circuit that is essential for both the establishment of fear memories and the expression of fear responses [13,14]. The posterior intralaminar nucleus of the thalamus has been studied in the context of pain induced by unconditioned stimuli, mainly because it projects to the basolateral amygdala [17].

The central medial nucleus of the thalamus (CMT), as a part of the intralaminar nucleus of the thalamus, is well-positioned to modulate fear-conditioning responses. The CMT has connections with two key structures that are important for fear behaviors—the prefrontal cortex and amygdala [18]. Additionally, the CMT is rich in Ca_V_3.1 T-type calcium channels (T-channels) and can modulate different behaviors through dual tonic and burst firing modes [19,20]. In comparison to high voltage-activated (HVA) calcium channels, T-channels need weaker depolarization to open and can form “window” currents around the resting neuronal membrane potential [21,22]. T-channels are de-inactivated during neuronal hyperpolarization, can be opened after depolarization, and are known to elicit low threshold spikes (LTSs) with characteristic rebound burst firing patterns [23]. Due to these unique properties, T-channels have an important role in regulating neuronal excitability, synaptic plasticity, and oscillatory behaviors [24,25]. T-channel dysfunction has been implicated in sleep disorders, anesthetic effects, the absence of epilepsy, pain, and neurological, neuropsychiatric, and cognitive disorders [22,26,27,28,29,30].

Recent studies have shown that pharmacological targeting of T-channels can also regulate the acquisition and recall of conditioned fear in animals [31,32]. In a genetic model with impaired fear extinction, enhanced burst firing in the thalamus was accompanied by increased T-currents, and pharmacological blockers of T-channels rescued the fear extinction deficiency [12]. These results suggest that T-channels may contribute to the neural systems that mediate the learning and memory of conditioned fear. However, the specific roles of the different thalamic T-channel isoforms in conditioned fear behavior are not clear. As we previously reported that Ca_V_3.1 T-channels are essential for CMT excitability [19,20], in this study, we set out to investigate the role of these channels in fear behavior using global knock-out (KO) and CMT-specific knock-down (KD) mice, thereby establishing thalamic T-channels as a potential novel target for the treatment of affective disorders.

## 2. Results

### 2.1. Contextual Fear Conditioning of Global Ca_V_3.1 KO Animals

In order to investigate the role of Ca_V_3.1 T-channels in fear memory processing, we first performed cued and contextual fear conditioning (CFC) experiments on control (WT) and global Ca_V_3.1 KO animals (Figure 1). A schematic of our fear-conditioning protocol is shown in Figure 1A. As thalamic Ca_V_3.1 T-channels have an important role in auditory processing [33], we only used two tone/shock pairings in our experiments, which resulted in less freezing related to the tone (Day 2) in comparison to the context (Day 3). Also, we did not investigate the role of thalamic T-channels on extinction processes in this experiment; therefore, further experiments are needed to explore this aspect.

Here, we found significantly higher freezing in Ca_V_3.1 KO animals during the training phase of the fear paradigm (Day 1), both over time (Figure 1B left) and over the whole training phase (Figure 1B right). In addition, although all the animals froze more after each tone/shock pairing (Figure 1C left), the mutant animals spent more time exhibiting freezing behaviors in comparison to WT mice at baseline (Figure 1C left, right) and had higher freezing percentages after each tone/shock pairing (Figure 1C left). Similarly, we found an increase in freezing behavior over time on Day 3 (testing phase, Figure 1D left) and an increase in the average freezing behavior in mutants in comparison to the control male mice (Figure 1D right).

### 2.2. Validation of AAV-shRNA Thalamic Injections

Next, we set out to investigate the specific role of Ca_V_3.1 channels in the CMT in fear behavior using the shRNA approach. To confirm the thalamic knock-down (KD) of Ca_V_3.1 T-channels (Figure 2A,B), we performed patch-clamp recordings using acute brain slices from wild-type (WT) mice injected with control (scrambled shRNA) or Cacna1g shRNAs (Figure 2C,D). Transfected neurons were readily identified in our recordings from live brain slices based on their high GFP immunofluorescence. Specifically, we compared the T-currents’ biophysical properties in GFP-positive neurons in the acute thalamic slices from control and Ca_V_3.1 KD animals. T-currents were evoked using our standard inactivation protocols with a hyperpolarizing step of −120 mV and a test potential of −50 mV (Figure 2C). Original traces of the inward calcium currents from representative recordings in the CMT in GFP-positive neurons from control animals (black traces) and Cacna1g shRNA-injected animals (green traces) are presented in Figure 2C. On average, we found that T-currents in GFP-positive neurons from Ca_V_3.1 KD mice (green symbols) were almost completely abolished compared to GFP-positive neurons from the control animals (gray symbols) (Figure 2D). In support of this finding, our qPCR data showed about a 76% reduction in Cacna1g RNA expression levels in CMT tissues from Cacna1g shRNA-injected animals in comparison to the control animals (Figure 2E).

Furthermore, we recorded current–voltage (IV) curves using a steady-state activation protocol to further validate the functional KD approach (Appendix A). Representative T-current traces from GFP-negative (black trace, control) and GFP-positive (green trace) neurons from Cacna1g shRNA-injected animals are presented in Appendix A. Our data showed a large reduction in the T-current amplitude over the range of the tested potentials (from −80 mV to −40 mV) in Cacna1g shRNA GFP-positive cells in comparison to GFP-negative (control) neurons in the CMT (Appendix A). In addition, the average voltage dependence of the steady-state activation (G/G_max_) curves showed a moderate right shift in the activation of T-current kinetics recorded from GFP-positive neurons, with a more positive mean value for V_50_ (−59.43 mV) when compared to the control (−65.58 mV, *p* > 0.05) (Appendix A).

### 2.3. Current-Clamp Experiments

The decreased T-current amplitudes strongly suggest that the excitability of the GFP-positive CMT neurons may be reduced following intra-thalamic injections of Cacna1g shRNA. Therefore, we performed current-clamp experiments and compared the tonic and burst-firing properties of GFP-positive neurons in the thalamus (Figure 3). Original traces of the tonic and rebound firing with the T-channel-dependent low threshold calcium spike (LTS) from a GFP-positive control neuron (black trace) and GFP-positive neuron from Ca_V_3.1 KD mice (green trace) are depicted in Figure 3A. We found that injections of Cacna1g shRNA into the CMT almost completely abolished the rebound burst firing mode and LTSs (green trace). In contrast, we found very little difference between the two groups when we compared the tonic firing in response to escalating depolarizing current injections from +25 to +275 pA (Figure 3B). Consequently, our analysis of the average number of rebound action potentials (APs) in GFP-positive neurons resulting from progressively stronger hyperpolarizing steps from -25 to -225 pA showed a profound reduction in rebound burst firing in Cacna1g shRNA-transfected thalamic neurons (Figure 3C). In addition, in GFP-positive thalamic neurons from Ca_V_3.1 KD animals, T-channel-dependent LTSs were completely absent in comparison to control GFP-positive neurons (Figure 3D). Interestingly, in a few GFP-positive Cacna1g shRNA-transfected thalamic neurons, we recorded APs after hyperpolarization that did not have characteristic LTSs and had a lower threshold for rebound AP firing (Figure 3E). We also found that the resting membrane potentials (RMPs) were similar between the two groups (Cacna1g shRNA: -59.95 *±* 0.61 mV; control shRNA: -59.96 *±* 0.53 mV (mean *±* SEM)), but the average input resistance was lower in GFP-positive CMT neurons from the Cacna1g shRNA group (Cacna1g shRNA: 134.2 *±* 18.7 MΩ; control shRNA: 273.6 *±* 46.4 MΩ (mean *±* SEM); unpaired two-tailed *t*-test t_10_ = 2.784, *p* = 0.02). However, it is unlikely that the changes in input resistance per se could cause the rebound burst-firing differences that we observed between the two groups, as their tonic firing modes were similar.

Overall, our ex vivo patch-clamp recordings of thalamic neurons in animals injected with Cacna1g shRNA showed that the GFP-positive neurons exhibited a strong decrease in T-current amplitudes and greatly diminished rebound burst firing in comparison to control GFP-positive neurons.

### 2.4. Contextual Fear Conditioning in CMT-Specific Ca_V_3.1 KD Animals

To investigate the specific contribution of thalamic Ca_V_3.1 T-channels to the observed changes in fear conditioning, we performed the same CFC protocol using CMT-specific Ca_V_3.1 KD and control mice (Figure 4A). In contrast to the global Ca_V_3.1 KO animals, the mice injected with Cacna1g shRNA in the CMT region showed very little difference in their freezing behavior during the training phase over time (Day 1, Figure 4B left) and over the whole acquisition phase (Day 1, Figure 4B right). Similar to global Ca_V_3.1 KO animals, there was higher freezing in all groups after each tone/shock pairing (Figure 4C left), but without any differences at baseline (Figure 4C left, right) or in freezing related to the tone/shock between the tested groups (Figure 4C left).

However, similar to Ca_V_3.1 KO mice, we found that thalamic Cacna1g shRNA mice exhibited higher freezing percentages during the testing phase over time (Day 3, Figure 4D left) and overall (Day 3, Figure 4D right). We concluded that in vivo silencing of Ca_V_3.1 channels in the CMT region affected the expression of freezing behaviors in the CFC paradigm, while global Ca_V_3.1 deletion affected both freezing during training (acquisition) and the expression of fear responses.

### 2.5. Cued Fear Conditioning in Global Ca_V_3.1 KO and CMT-Specific Ca_V_3.1 KD Animals

To study the role of thalamic Ca_V_3.1 T-channels in cue-related fear responses, we analyzed the freezing behavior related to tones in the different contexts (Day 2) in global Ca_V_3.1 KO and thalamus-specific Ca_V_3.1 KD mice (protocol shown in Figure 5A). We observed that WT and Ca_V_3.1 KO mice froze more after each tone, without any differences in the fear responses between the groups over time (Figure 5B left) or after each tone pairing (Figure 5B right). The same effect was observed in Cacna1g shRNA and control animals (Figure 5C). Both tested groups had higher freezing percentages after each tone over time (Figure 5C left) and after each tone pairing (Figure 5C right). There were no differences between the groups in terms of freezing behavior. As we have previously detected higher baseline freezing in Ca_V_3.1 KO animals on Day 1 (Figure 3C right), we tested the baseline freezing in Ca_V_3.1 KO and WT mice on Day 2 and observed a similar trend (unpaired *t*-test for baseline freezing percentage *p* = 0.06).

### 2.6. General Activity and Anxiety-Related Behaviors

Additionally, we examined the effect of a targeted reduction in Ca_V_3.1 T-channel function in the thalamus on general activity and anxiety-related behaviors using open-field (Figure 6A) and zero-maze (Figure 6E) tests. We found that the performance of the mice injected with either scrambled or Cacna1g shRNA did not differ in the open-field paradigm, as assessed by the time in the central zone (Figure 6B), the number of entries into the central zone (Figure 6C), or the total distance traveled (Figure 6D). Similarly, we did not observe differences in the zero-maze behavior between the tested groups in terms of the time spent in the open quadrants (Figure 6F), the number of entries to the open quadrants (Figure 6G), or the distance traveled (Figure 6H). Hence, it appears that the intra-thalamic injection of Cacna1g shRNA did not alter the anxiety-related behaviors or general motor abilities of the mice.

## 3. Discussion

In this study, we demonstrated for the first time that by silencing thalamic CMT Ca_V_3.1 T-channels, we can modulate fear expression, thus suggesting that Ca_V_3.1 T-channels could be a potential cellular target for disorders where there is an altered fear response such as post-traumatic stress disorder, schizophrenia, anxiety, or substance abuse.

In the context of fear conditioning, the thalamus is considered to play a crucial role as a sensory relay station, receiving sensory information from the periphery and transmitting it to the amygdala, which is the primary brain region responsible for processing fear. In addition, recent studies have implicated non-sensory parts of the thalamus (midline and intralaminar nuclei of the thalamus) in fear-conditioning behaviors [13,15,16,17]. The midline and intralaminar thalamic nuclei are well-placed to contribute to fear behaviors due to the extensive projections that they receive from the periaqueductal gray while projecting to the prefrontal cortex and amygdala [17,34]. In addition, most of the midline and intralaminar thalamic nuclei are part of the thalamo-hippocampal circuitry and can therefore influence memory-related behaviors [35]. Specifically, animal studies have shown that the PV regulates fear processing through selective inactivation of the lateral division of the central amygdala [13] and by tuning the inhibitory functions in the prefrontal cortex [15,16]. Additionally, conditioned stimulus–unconditioned stimulus presentations caused increased c-Fos expression in the prefrontal cortex, midline, and intralaminar thalamus (including the PV and CMT), the lateral amygdala, and retrograde-labeled midline thalamic afferents to the prefrontal cortex [13,36].

Fear conditioning is one of the most frequently used experimental procedures not only for modeling learning and memory but also for anxiety disorders. The conditional freezing response is used as the output for most fear-conditioning experiments in rodents. Here, we showed that global Ca_V_3.1 KO animals froze more at baseline and immediately after each tone/shock pairing in the acquisition–training phase. The higher baseline freezing in Ca_V_3.1 KO mice implies that these mutant animals have different emotional responses to novel and non-aversive environments, at least in the first few minutes. This baseline difference in freezing can partially explain the stronger freezing behavior related to tones/shocks observed in the mutant animals. Additionally, there is a possibility that the brief increase in freezing of global KO animals after each tone/shock pairing was due to the known role of thalamic Ca_V_3.1 T-channels in auditory processing [33]. Ca_V_3.1 T-type calcium channels play a significant role in auditory perception. They contribute to the electrical activity of neurons in the auditory pathway, particularly in the thalamus, which is crucial for regulating the firing pattern of neurons, impacting how sound information is transmitted to the auditory cortex. Hence, they can potentially influence aspects like sound localization and the perception of sound intensity [33]. It has been shown that the switch between firing modes depends on thalamic Ca_v_3.1 T-type calcium channels and that pharmacologic and/or genetic inhibition of these channels in the auditory thalamus substantially influences auditory processing [33]. On the other hand, based on our findings, we propose that an increase in freezing behavior after each tone/shock pairing was not due to different pain thresholds as our data showed no pain threshold differences between WT and Ca_V_3.1 KO animals, but it could be present due to an altered emotional processing of pain.

As Ca_V_3.1 T-channels are not only abundantly expressed in the thalamus but also in the cortex and other brain regions important for pain processing and modulation [37], it is not surprising to see different fear responses in global Ca_V_3.1 KO animals after aversive stimuli, i.e., control (WT) mice exhibited normal freezing behaviors when re-exposed to the context associated with the aversive stimulus, whereas Ca_V_3.1 KO mice had stronger freezing responses, suggesting the existence of altered fear expression in these mutant animals. Similarly, CMT-specific Ca_V_3.1 KD animals spent more time freezing than the control animals during the testing phase, emphasizing the important role of thalamic Ca_V_3.1 T-channels in fear expression.

Previously, we showed that the Ca_V_3.1 T-channel subtype is the dominant isoform in the CMT and that CMT neurons from Ca_V_3.1 KO animals have a reduced ability to burst fire [20]. Ca_V_3.1 T-channels are important in the burst-firing mode of thalamic neurons [19,20], and here, we were able to modulate the fear-related responses in Ca_V_3.1 KD animals by reducing T-currents and diminishing the ability of CMT neurons to fire in the burst mode. Interestingly, CMT-specific Ca_V_3.1 T-channel silencing did not affect the fear behaviors during training, as was seen with the global Ca_V_3.1 KO animals, suggesting a more selective effect of thalamic Ca_V_3.1 T-channels on fear expression. Our data are supported by previous findings reporting that the injection of a T-channel blocker before the acquisition of fear conditioning significantly increased freezing in rats [31]. These authors commented that the enhanced fear behavior observed prior to acquisition and extinction, as well as during acquisition, could reflect an increase in anxiety after the systemic T-channel blocker injection [31]. This could also explain the enhanced fear behavior we observed in Ca_V_3.1 KO mice during acquisition and fear expression. However, although the animals with global deletion of Ca_V_3.1 T-channels may have hippocampal-dependent learning and memory deficits, the general locomotor activity and anxiety-related behaviors of Ca_V_3.1 KO male mice were unchanged [38]. Additionally, in our experiments, we did not observe changes in the locomotor activity and anxiety-related parameters of the thalamic Ca_V_3.1 KD animals in comparison to the control mice, suggesting that the enhanced freezing behaviors during testing cannot be attributed to an increase in anxiety in these animals.

As thalamic Ca_V_3.1 T-channels are important for auditory processing [33], we used a protocol with a reduced number of tone/shock pairings in CFC experiments. The CFC behavioral paradigms with five or more tone/shock pairings typically have stronger and persistent fear memory, often accompanied by enhanced generalization to similar cues and contexts [39]. In contrast, two pairings, as in our experiment, usually produce weaker fear responses, which facilitate the assessment of subtler cognitive processes such as discrimination, extinction, or precise neural circuit involvement [39]. Thus, the number of pairings directly influences the intensity and stability of conditioned fear responses and may impact both behavioral outcomes and neurobiological mechanisms studied. Although in our experiments all the animals had a higher freezing percentage after each tone in a different context (Day 2), we did not observe differences in freezing between the groups related to the tone. This suggests that Ca_V_3.1 T-channels may be more important for contextual conditioning to an unpleasant stimulus in comparison to the tone-related fear or that our conditioning to the cue was too subtle. As this could be a limitation of our study, further experiments are needed to fully investigate the role of CMT Ca_V_3.1 T-channels on cue-related fear conditioning and the processing of fear extinction.

It is known that the shRNA approach, while effective for gene knock-down, may have potential off-target effects [40]. However, in our experiments, knocking down Ca_V_3.1 T-channels only affected rebound burst firing but not tonic firing. This strongly suggests that, as we used the shRNA approach, we did not significantly affect other major targets that control neuronal excitability. Additionally, our experimental method is justified as the use of shRNA to manipulate genes in different tissues is a commonly used approach in animal studies and shRNA has enormous potential as a precision-based therapy to treat numerous neurodegenerative disorders [41,42].

Using CMT-specific Ca_V_3.1 KD mice, we found that fear expression can be modulated by reducing the ability of thalamic CMT neurons to fire in burst mode. Similarly, it has been reported that the firing mode of the MD thalamus is critical for the modulation of fear extinction [12]. These authors showed that the mutant mice exhibited impaired fear extinction, accompanied by enhanced burst firing and increased T-currents in MD neurons. Importantly, the abnormal fear extinction was rescued by pharmacologically blocking T-channels in vivo [12]. Together with our data, these findings highlight the key role of midline and intralaminar thalamic T-channels in modulating fear expression and extinction. We conclude that, through regulating T-channel activity, we can modulate the firing mode of thalamic neurons and consequently affect fear-mediated responses.

## 4. Materials and Methods

### 4.1. Animals

The experimental procedures with animals were performed according to the guidelines approved by the University of Colorado Anschutz Medical Campus. All efforts were made to minimize animal suffering and to only use the number of animals necessary to produce reliable scientific data. All experiments were conducted during the light cycle in male adult (2–5-month-old) C57BL/6J wild-type (WT) and Ca_V_3.1 knock-out (KO; Ricken BioResources Centre, Japan) mice, as it is well-documented that the estrous cycle can have a significant effect on associative learning [43]. All researchers were blinded during the behavioral and ex vivo electrophysiological experiments.

### 4.2. Generation of CMT-Specific Ca_V_3.1 Knock-Down (KD) in Mice

The use of global Ca_V_3.1 knock-out (KO) mice is useful for proof-of-principle confirmation, but the widespread distribution of these channels and possible compensatory modulations is a major disadvantage of this approach. To specifically investigate the contribution of Ca_V_3.1 T-channels in the thalamus, we used mice with CMT-specific silencing of the Cacna1g gene, which encodes the Ca_V_3.1 channel. Our method consisted of knocking down Cacna1g by injecting short hairpin RNA (shRNA) using a procedure described elsewhere [38,44]. Adult male C57BL/6J mice were anesthetized with 1–2% isoflurane, given 2.5 L/min of oxygen, and transferred to a standard stereotaxic frame (Kopf Instruments, CA, USA). The mice were randomized into two treatment groups: scrambled control (AAV2-GFP-U6-scrmb-shRNA; titer: 1.1 × 10^13^ GC/mL; Vector Biolabs, Malvern, PA) or Cacna1g shRNA targeting Ca_V_3.1 (AAV2-GFP-U6-mCACNA1G-shRNA; titer: 6.8 × 10^12^ GC/mL; Vector Biolabs) groups. The shRNAs were injected into the CMT [anteroposterior (AP): −1.35 mm; mediolateral (MD): 0; and dorsoventral (DV): −3.6 mm] using high-titer AAV2 (0.5 μL, using a 5 μL Hamilton syringe at a rate of 0.1 µL/min). The animals were allowed to recover for at least three weeks before the experiments (behavioral or ex vivo brain slice recordings) were performed to ensure adequate virus expression. Although we aimed to target just CMT neurons in our experiments, we are aware that other parts of the thalamus could have been partially transfected (mostly the IMD, PV, and MD; Figure 1B). We excluded all animals that did not have the highest Cacna1g shRNA or scrambled shRNA expression in GFP-positive neurons in the CMT in comparison to other areas. The WT animals were randomly assigned to the different groups such that animals from the same litter were used for the generation of both control and Ca_V_3.1 KD animals. Animals from at least 3 different litters were used for the experiments.

### 4.3. Ex Vivo Brain Slice Preparation

The mice were briefly anesthetized with 5% isoflurane and decapitated. Their brains were quickly removed and placed in a cold (4 °C) oxygenated (95 vol% O_2_ and 5 vol% CO_2_) solution. Live 250 μm thick coronal brain slices were sectioned at 4 °C using a vibrating microslicer in a cold solution containing 260 mM sucrose, 10 mM D-glucose, 26 mM NaHCO_3_, 1.25 mM NaH_2_PO_4_, 3 mM KCl, 2 mM CaCl_2_, and 2 mM MgCl_2_ (Laica VT 1200S). The brain slices were immediately incubated for 30 min in a solution containing the following at 37 °C: 124 M NaCl, 10 mM D-glucose, 26 mM NaHCO_3_, 1.25 mM NaH_2_PO_4_, 4 mM KCl, 2 mM CaCl_2_, and 2 mM MgCl_2_. The electrophysiology experiments were carried out at room temperature. During the incubation, the slices were constantly perfused with a gas mixture of 95 vol% O_2_ and 5 vol% CO_2_.

Whole-cell recordings of CMT neurons were performed using Zeiss optics (Zeiss AXIO Examiner D1, ×40 objective). Glass microelectrodes (Sutter Instruments, borosilicate glass with filament OD of 1.2 mm) were pulled using a Sutter Instruments P-1000 model and fabricated to maintain an initial resistance of 4–6 mΩ. Neuronal membrane responses were recorded using a Multiclamp 700 B amplifier (Molecular Devices, Foster City, CA, USA). The voltage–current commands and the digitization of the resulting voltages and currents were performed using Clampex 8.3 software (Molecular Devices) running on a PC-compatible computer. The resulting current traces were analyzed using Clampfit 10.5 (Molecular Devices). The statistical and graphical analyses were performed using GraphPad Prism 9.0 (GraphPad Software) or Origin 7.0 (OriginLab) software. The results are presented as the mean ± SEM unless stated otherwise.

### 4.4. Electrophysiology Experiments Using Brain Slices

The external solution for current-clamp and voltage-clamp electrophysiology experiments consisted of the following: 125 mM NaCl, 25 mM D-glucose, 25 mM NaHCO_3_, 1.25 mM NaH_2_PO_4_, 2.5 mM KCl, 1 mM MgCl_2_, and 2 mM CaCl_2_. For the current-clamp experiments, the external solution contained the synaptic blockers picrotoxin (20 μM), D-2-amino-5-phosphonovalerate (D-AP5; 50 µM), and 2,3-dihydroxy-6-nitro-7-sulfamoyl-benzo [f]quinoxaline-2,3-dione (NBQX; 5 µM). For the voltage-clamp experiments measuring T-type calcium channels, tetrodotoxin (TTX; 1μM) was added to the extracellular medium as a blocker of voltage-dependent sodium currents. The internal solution for the current-clamp recordings consisted of the following: 130 mM potassium-D-gluconate, 5 mM ethylene-glycol-bis (β-aminoethylether)*N,N,N′,N′*-tetra acetic acid (EGTA), 4 mM NaCl, 0.5 mM CaCl_2_, 10 mM HEPES, 2 mM Mg-ATP, and 0.5 mM Tris-GTP, pH 7.2. For the T-channel recordings, we used an intracellular cesium-based internal solution containing the following: 110 mM Cs-methanesulfonate, 14 mM phosphocreatine, 10 mM HEPES, 9 mM EGTA, 5 mM Mg-ATP, and 0.3 mM Tris-GTP, with the pH adjusted to 7.15–7.20 using CsOH (standard osmolarity: 300 mOsm) [19]. For the voltage-clamp experiments, maximal inactivation curves were generated using a standard double-pulse protocol with 3.6 s long prepulses (voltage: −120 mV) and test potentials (−50 mV). For additional voltage-clamp experiments, activation (IV) curves were generated using a standard protocol where the holding potential (Vh) of −90 mV was increased to depolarized test potentials (Vt) from −80 to −40 mV in 5 mV increments. The voltage dependencies of activation were described using a single Boltzmann distribution: G (V) = G_max_**/** (1 + exp [-(V − V_50_)**/***k*]), where G_max_ is the maximal conductance (calculated by dividing the current amplitude by the estimated reversal potential), V_50_ is the voltage at which half of the current is activated, and *k* represents the voltage dependence (slope) of the distribution. The average current amplitudes were analyzed using the appropriate statistics.

In the current clamp experiments, both the stimulated tonic and burst-firing properties of the CMT neurons were characterized using multistep protocols. To investigate the stimulated tonic firing patterns in CMT cells, we injected a depolarizing current pulse through the recording pipette with a 400 ms duration in 25 pA increments, starting from 50 pA. To investigate the rebound burst-firing patterns, the neurons were injected with hyperpolarizing currents in 25 pA intervals from 0 to −225 pA. The subsequently stimulated tonic action potential (AP) frequencies, rebound burst-firing thresholds, and APs in the rebound burst were determined. The resting membrane potential was measured at the beginning of each recording and the liquid junction potential was not corrected. All recordings were obtained using a −60 mV membrane potential.

### 4.5. Quantitative Real-Time PCR

After anesthetizing the animals with isoflurane and removing the brain, each brain was sliced on a vibratome (Leica VT 1200S) in ice-cold PBS, creating 500 μm thick horizontal slices. CMT tissue punches with GFP-positive labeled neurons were collected under a dissecting fluorescence microscope. Samples from 3 animals from the Cacna1g shRNA group and 4 animals from the control (scrambled shRNA) group were pooled and the samples were stored at -80 °C until further use. RNA was isolated using the RNeasy Microarray Tissue Mini Kit with QiAzol (QIAGEN), and quantitative real-time reverse transcription (qRT)–PCR was performed on a BioRad Icycler using the RT2 First Strand Kit and RT2 SYBR Green qPCR Mastermix (QIAGEN) according to the manufacturer’s protocols. The primers for CaV3.1 and cyclofilin (internal control) were purchased from QIAGEN (CACNA1G NM_031601 catalog number: PPR52633A-200; cyclophilin NM_001004279 catalog number: PPR59729A). The qRT-PCR data were analyzed as previously described in [45].

### 4.6. Behavioral Experiments

The investigators were blinded to the genotype in all the behavioral experiments. All tests were performed by trained researchers and monitored and analyzed using specialized software (Ethovision XT system, Noldus, Wageningen, The Netherlands). The behavioral experiments involving virus-injected mice began at least three weeks after injection to allow for adequate expression levels. The animals were habituated to the testing room for at least 30 min before all behavioral experiments.

For the contextual fear conditioning test, the mice were placed into conditioning boxes that were 30.5 cm × 24.1 cm × 29.2 cm in size. Inside the box was a metal grid floor, a house light, and an odor cue (70% ethanol, which was also used to clean the apparatus). On the first day of training, the mice were placed in the chamber for 7 min, during which 2 tones (30 s in duration) paired with shocks (1 s, 0.7 mA) were presented. On the second day, the mice were placed in a different environment for 7 min, and just the tone (30 s, 2 times) was present without a shock. On the third day, the mice were placed into the same box as Day 1 (same box, but no shock or tone applied), and the mouse’s behavior was monitored for 7 min. Fear behavior was analyzed and presented as the percent of time spent in a freezing state. The conditioning boxes were cleaned with ethanol after each animal, making sure that the ethanol had evaporated completely before placing a new mouse in the chamber. The cleaning was performed the same way during the experiments (Days 1–3) so that the odor cue was always constant.

For the open-field test, the mice were placed in the center of the arena (gray box, 44 cm × 44 cm × 30 cm) and tested for 10 min without prior habituation. This test provided important information about the general locomotor activity (total distance traveled), as well as the anxiety-like behaviors (number of entries and time spent in the center of the arena) of the mice.

The elevated zero maze is a modification of the elevated Plus Maze test and it is used to investigate anxiety-like behaviors in mice. It has an elevated ring-shaped runway (50 cm diameter, 5 cm width, and 20 cm wall height) with equal areas for the adjacent open and closed quadrants. The animals were placed in the center of the open quadrant and tested for 10 min. Similar to the open-field test, the test gives information about general locomotor activity (total distance traveled), as well as anxiety-like behaviors (number of entries and time spent in the center quadrat) of mice.

The open-field and zero-maze apparatuses were cleaned with ethanol after each animal, making sure that the ethanol had evaporated completely before placing a new mouse in the chamber. Cleaning was performed the same way during all the behavioral experiments.

### 4.7. Data Analysis

In every ex vivo electrophysiological experiment, we attempted to obtain as many recordings as possible from the neurons to minimize the number of animals used. The statistical analysis was performed using two-way repeated measure (RM) ANOVA, as well as Student’s unpaired two-tailed *t*-test where appropriate. For significant interactions between factors according to two-way RM ANOVA, Sidak’s post hoc comparisons were used. *p* values < 0.05 were considered significant. Additionally, Cohen’s *d* and effect size *r* were reported. The statistical and graphical analyses (presented as the mean ± standard error of the mean) were performed using the GraphPad Prism 10 software (GraphPad Software, La Jolla, CA, USA) and the traces were prepared using Origin 2018 (OriginLab, Northampton, MA, USA). The schematic presentations of the CFC, open-field, and zero-maze tests were created using bioRender (BioRender.com).

## 5. Conclusions

In conclusion, we reported that thalamic Ca_V_3.1 T-type calcium channels are a crucial regulator of fear behavior in mice. Considering the significant role of the midline and intralaminar nuclei of the thalamus in fear learning and responses to aversive stimuli, our data indicate that Ca_V_3.1 T-channels may be a promising novel drug target for treating altered fear responses associated with multiple brain disorders.

## Figures and Tables

**Figure 1 ijms-26-03543-f001:**
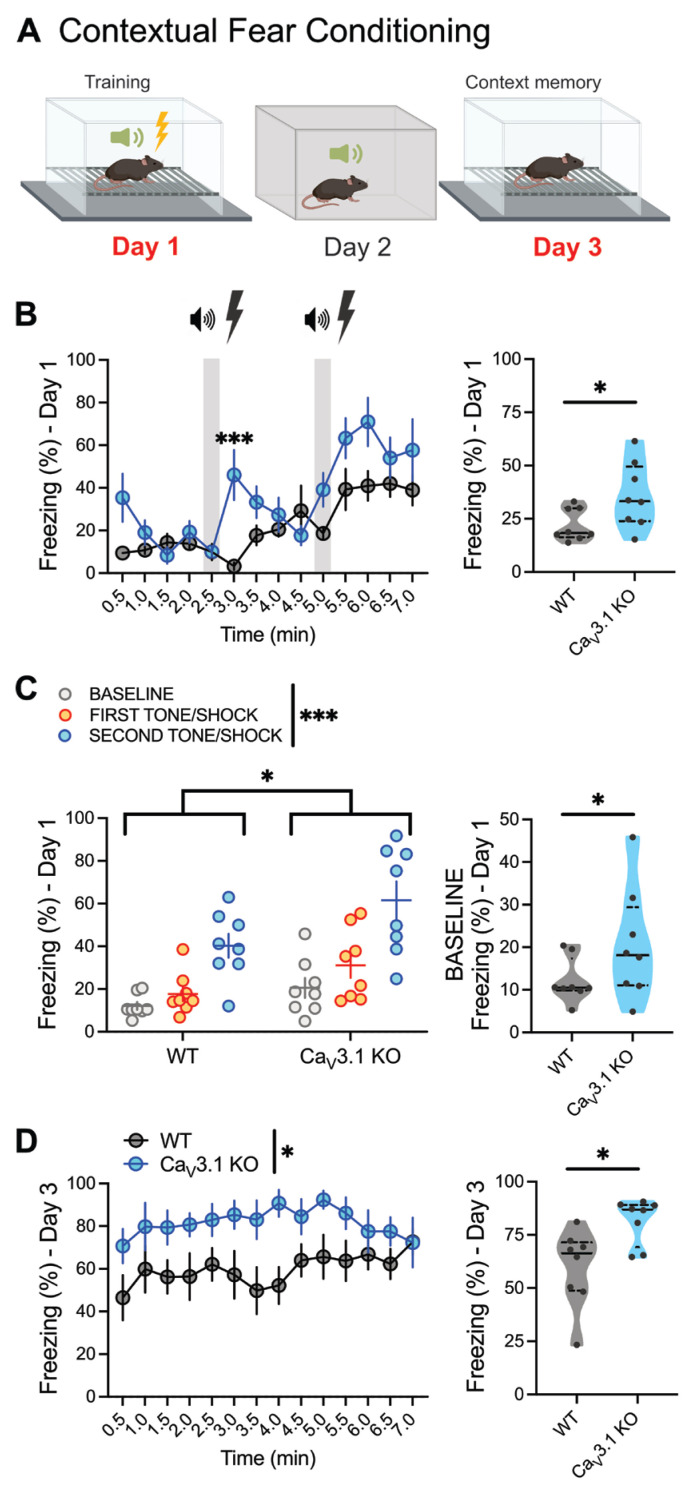
Global deletion of Ca_V_3.1 T-channels increases fear responses during acquisition and expression of contextual fear conditioning. (**A**) Schematic diagram of contextual fear conditioning: Day 1—training (acquisition); Day 2—different context; Day 3—testing. (**B**) Percentage of time spent freezing in control wild-type (WT, gray) and Ca_V_3.1 knock-out (KO, blue) mice during training phase of contextual fear conditioning paradigm, analyzed every 30 s (left) and over whole training phase (right). Ca_V_3.1 KO animals showed amplified freezing behaviors after first shock (left; two-way RM ANOVA: interaction F_(13,182)_ = 2.767, *p* = 0.001; time F_(13,182)_ = 12.32, *p* < 0.001; genotype F_(1,14)_ = 5.138, *p* = 0.04; Sidak’s post hoc test) and spent more time freezing overall during 7 min acquisition phase (right; unpaired two-tailed *t*-test: t_(14)_ = 2.27, *p* = 0.040; Cohen’s *d* = 1.13; effect size *r* = 0.49). (**C**) Although there was increased freezing after each tone/shock pairing in WT and mutant animals, Ca_V_3.1 KO mice had a higher overall freezing percentage than controls (left, two-way RM ANOVA: interaction F_(2,28)_ = 1.587, *p* = 0.222; tone F_(2,28)_ = 49.75, *p* < 0.001; genotype F_(1,14)_ = 4.810, *p* = 0.046). Baseline freezing behavior was higher in global Ca_V_3.1 KO male mice in comparison to WT animals (right; unpaired two-tailed *t*-test: t_(14)_ = 1.688, *p* = 0.026; Cohen’s *d* = 0.844; effect size *r* = 0.39). (**D**) Compared to WT mice, percentage of time freezing of Ca_V_3.1 KO mice during testing significantly increased over time (left; two-way RM ANOVA: interaction F_(13,182)_ = 0.867, *p* = 0.589; time F_(13,182)_ = 0.875, *p* = 0.581; genotype F_(1,14)_ = 7.688, *p* = 0.011) and over whole 7 min testing period (right; unpaired two-tailed *t*-test: t_(14)_ = 2.94, *p* = 0.011; Cohen’s *d* = 1.47; effect size *r* = 0.59). *n* = 8 animals per group; * *p* < 0.05, *** *p* < 0.001.

**Figure 2 ijms-26-03543-f002:**
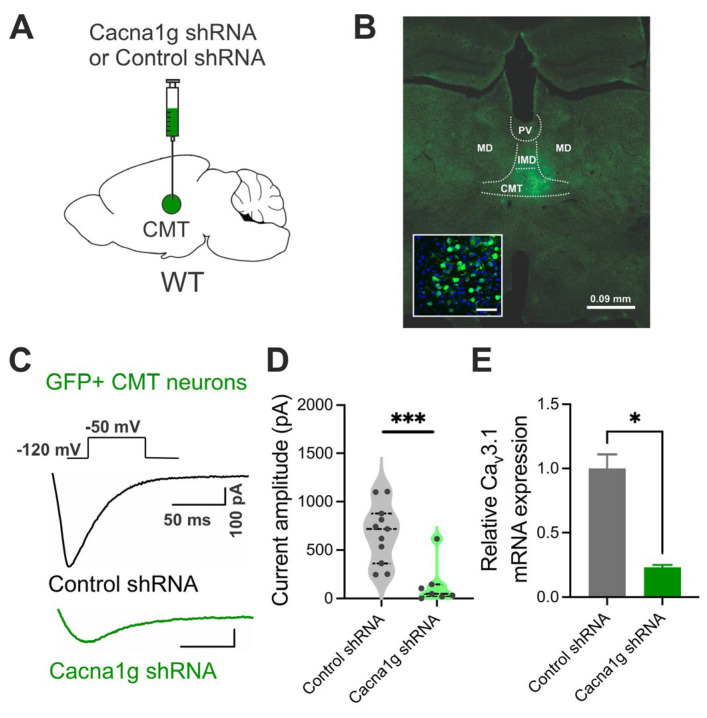
Initial validation of AAV-shRNA injections. (**A**) Schematic of CMT-specific knock-down (KD) animal generation by injecting Cacna1g shRNA (or scrambled shRNA as control). (**B**) Image shows GFP-transfected neurons only, counter-stained with DAPI. (**C**) T-current traces from GFP-positive neurons from animals injected with scrambled shRNA (control, black trace) or Cacna1g shRNA (green trace). (**D**) Average T-current amplitudes in GFP-positive neurons after control and Cacna1g shRNA injections (unpaired two-tailed *t*-test: t_(16)_ = 4.023, *p* < 0.001; Cohen’s *d* = −2.02; effect size *r* = -0.71). *n* = 11 control and 7 Cacna1g shRNA-transfected GFP-positive neurons; *** *p* < 0.001. (**E**) q-PCR data pooled from the tissues of 3 Cacna1g shRNA and 4 control animals showing 76.72% decrease in relative Ca_V_3.1 expression in Cacna1g shRNA group in comparison to control (unpaired two-tailed *t*-test: t_(2)_ = 6.893, *p* = 0.02; Cohen’s *d* = −6.89; effect size *r* = −0.96). *n* = 2, technical replicates; * *p* < 0.05.

**Figure 3 ijms-26-03543-f003:**
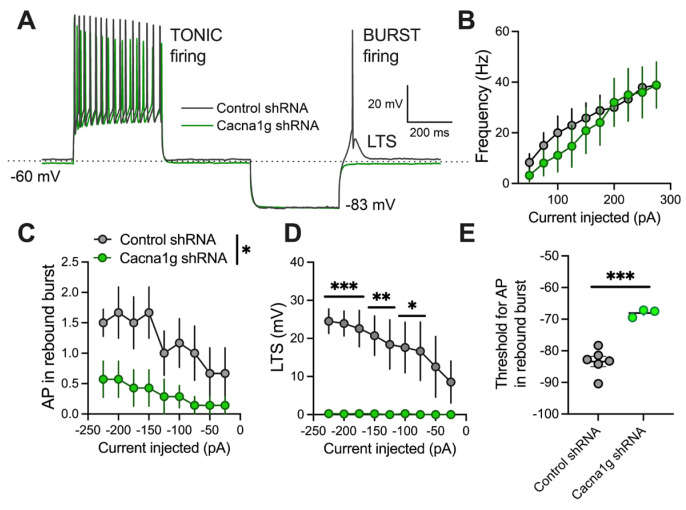
Different firing properties of GFP-positive thalamic neurons from control (scrambled shRNA) and Cacna1g shRNA-injected animals. (**A**) Original traces from representative thalamic neurons from GFP-positive control (black) and Cacna1g shRNA-transfected (green) neurons showing active membrane responses to depolarizing (275 pA) and hyperpolarizing (−225 pA) current injections. Note that GFP-positive neurons from Cacna1g shRNA cohort did not show action potentials (APs) with low threshold spikes (LTSs) after membrane hyperpolarization. (**B**) There was no difference in average tonic AP firing frequency (50–275 pA current injections) between GFP-positive thalamic neurons from control and Cacna1g shRNA mice. (**C**) Number of APs in rebound burst was statistically significantly smaller in GFP-positive neurons from Cacna1g shRNA animals (two-way RM ANOVA: interaction F_(8,88)_ = 0.963, *p* = 0.470; injected current F_(8,88)_ = 4.92, *p* < 0.001; GFP F_(1,11)_ = 5.133, *p* = 0.045). (**D**) LTSs were not observed in thalamic GFP-positive Cacna1g shRNA-transfected neurons (two-way RM ANOVA: interaction F_(8,88)_ = 3.50, *p* = 0.001; injected current F_(8,88)_ = 3.748, *p* < 0.001; GFP F_(1,11)_ = 15.02, *p* = 0.003; Sidak’s post hoc test). (**E**) Threshold for rebound burst firing was higher in thalamic neurons from Cacna1g shRNA-injected animals (unpaired two-tailed *t*-test: t_(7)_ = 6.314, *p* < 0.001). Note that this rebound firing had APs without expected T-channel-dependent LTSs. *n* = 6 control and 7 Cacna1g shRNA-transfected GFP-positive neurons; * *p* < 0.05, ** *p* < 0.01, *** *p* < 0.001.

**Figure 4 ijms-26-03543-f004:**
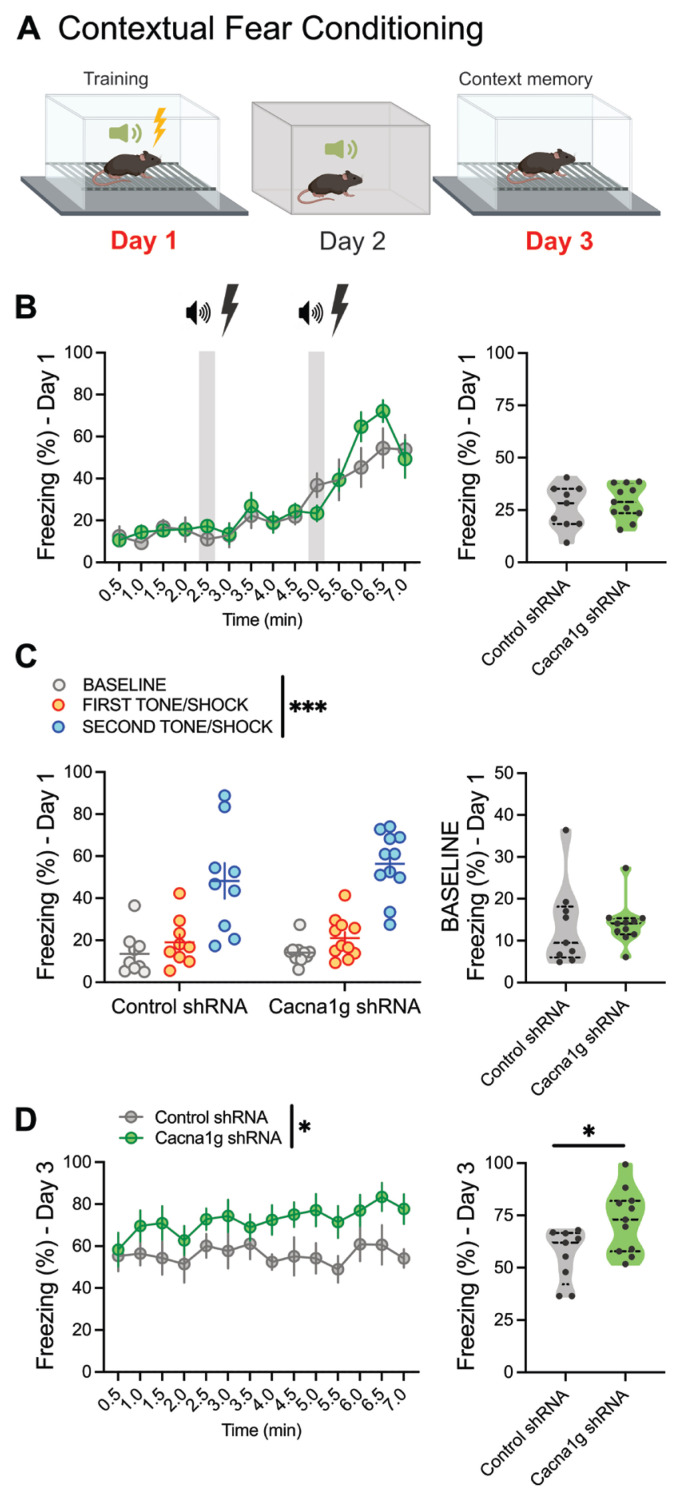
Targeted reduction in thalamic Ca_V_3.1 T-channels increases fear responses during expression but not during acquisition of contextual fear conditioning. (**A**) Schematic diagram of contextual fear conditioning: Day 1—training (acquisition); Day 2—different context; Day 3—testing. (**B**) Percentage of time spent freezing in control (scrambled shRNA, gray) and Cacna1g shRNA-injected (blue) mice during training phase of contextual fear conditioning paradigm, analyzed every 30 s (left) and over whole training phase (right). (**C**) Increased freezing after each tone/shock pairing in control and Ca_V_3.1 KD mice without any differences in freezing behavior between groups (left; two-way RM ANOVA: interaction F_(2,36)_ = 0.182, *p* = 0.834; tone F_(2,36)_ = 23.56, *p* < 0.001; shRNA F_(1,18)_ = 0.090, *p* = 0.768). Control and Cacna1g shRNA-injected animals had similar baseline freezing percentages (right). (**D**) Compared to control mice, percentage of time freezing in Cacna1g shRNA-injected mice during testing significantly increased over time (left; two-way RM ANOVA: interaction F_(13,234)_ = 0.551, *p* = 0.891; time F_(13,234)_ = 0.991, *p* = 0.461; shRNA F_(1,18)_ = 6.642, *p* = 0.019) and over whole 7 min testing period (right; unpaired two-tailed *t*-test: t_(18)_ = 2.577, *p* = 0.019; Cohen’s *d* = 1.17; effect size *r* = 0.50). *n* = 9 control and 11 Cacna1g shRNA-injected mice; * *p* < 0.05, *** *p* < 0.001.

**Figure 5 ijms-26-03543-f005:**
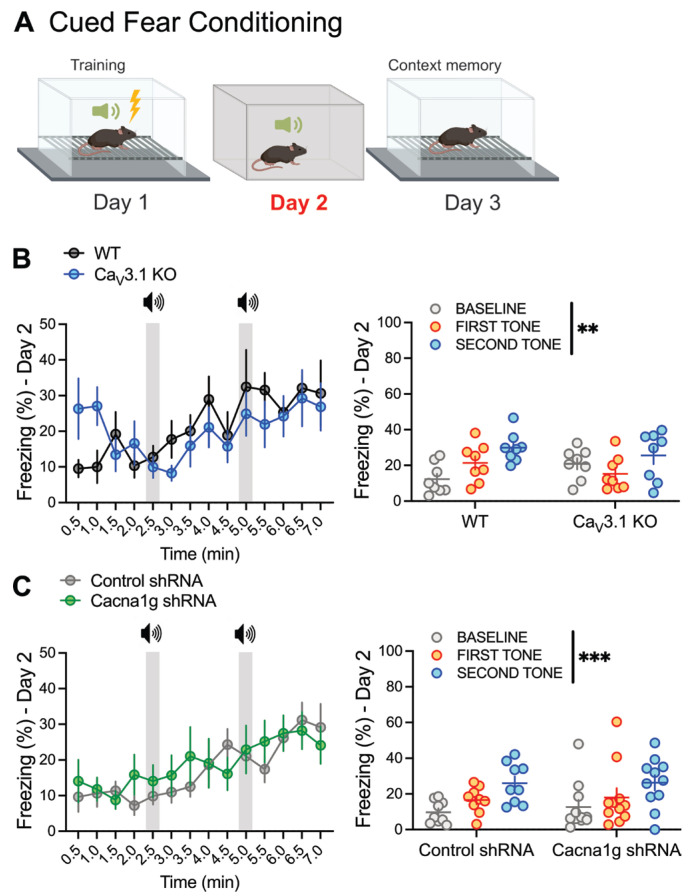
Global deletion and targeted reduction in thalamic Ca_V_3.1 T-channels did not change fear responses during cued fear conditioning. (**A**) Schematic of cued fear conditioning: Day 1—training (acquisition); Day 2—different context; Day 3—testing. (**B**) Percentage of time spent freezing in control (WT, gray) and Ca_V_3.1 KO (blue) mice during cued fear conditioning paradigm, analyzed every 30 s (left; two-way RM ANOVA: interaction F_(13,182)_ = 1.249, *p* = 0.248; time F_(13,182)_ = 2.791, *p* = 0.001; shRNA F_(1,14)_ = 0.156, *p* = 0.698), and average freezing responses at baseline and after each tone (right; two-way RM ANOVA: interaction F_(2,28)_ = 2.86, *p* = 0.074; tone F_(2,28)_ = 6.39, *p* = 0.005; shRNA F_(1,14)_ = 0.039, *p* = 0.846). (**C**) Percentage of time freezing in control (gray) and Cacna1g shRNA-injected mice (green) during cued fear conditioning, analyzed over time (left; two-way RM ANOVA: interaction F_(13,234)_ = 0.919, *p* = 0.534; time F_(13,234)_ = 6.164, *p* < 0.001; shRNA F_(1,18)_ = 0.124, *p* = 0.729) and averaged (right; two-way RM ANOVA: interaction F_(2,36)_ = 0.182, *p* = 0.834; tone F_(2,36)_ = 23.56, *p* < 0.001; shRNA F_(1,18)_ = 0.090, *p* = 0.768). *n* = 8 WT, 8 Ca_V_3.1 KO, 9 control, and 11 Cacna1g shRNA-injected mice; ** *p* < 0.01, *** *p* < 0.001.

**Figure 6 ijms-26-03543-f006:**
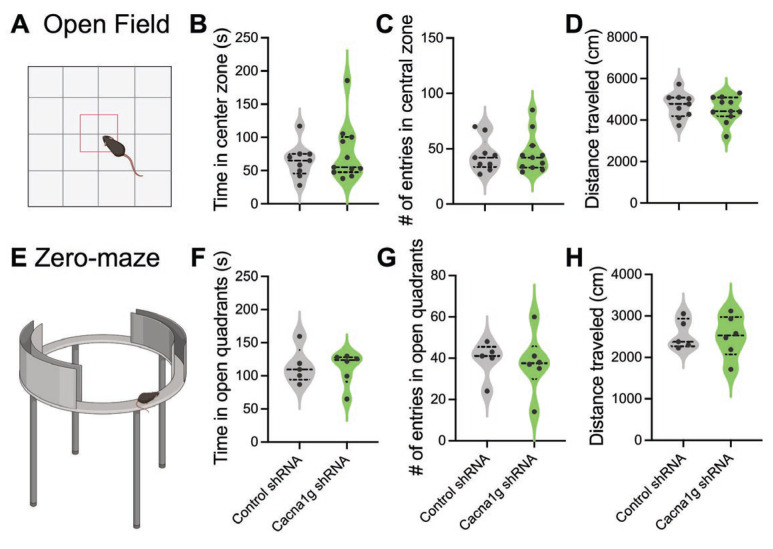
General activity and anxiety-related behaviors in Cacna1g shRNA-injected mice. (**A**) Graphic presentation of open-field test. (**B**) Time spent in the central zone of open-field arena. (**C**) Number of entries to central zone of open-field arena. (**D**) Distance traveled in open-field arena in control and Cacna1g shRNA mice. (**E**) Graphic presentation of zero-maze test. (**F**) Time spent in open quadrants. (**G**) Number of entries into open quadrants. (**H**) Distance traveled during zero-maze test in control and Cacna1g shRNA mice.

## Data Availability

Data will be made available upon request to the corresponding author.

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
