# Peer review of "Role of Thalamic CaV3.1 T-Channels in Fear Conditioning"

_ijms, 2025, doi:10.3390/ijms26083543_

Round 1

Reviewer 1 Report

Comments and Suggestions for Authors

This article describes an attempt by Stamenic TT et al to link behavioral changes with their previous findings that t-type Ca channels control excitability in the central thalamus. They use ex vivo electrophysiology and whole animal behavior protocols with an overall goal of an increased understanding of the mechanisms of fear conditioning and extinction in mammals. While the thalamus has been implicated in the process of fear memory consolidation, there is a paucity of investigations regarding mechanisms.

In this investigation, Stamenic TT find that knockdown of T-type calcium channels does not affect tonic firing of action potentials, but does reduce burst firing and low-threshold calcium spikes. In their behavior experiments they find that both global KO and targeted KD of t-channels increase in freezing during the testing phase. Their were curious differences in freeze response during the training phase that can likely be explained by incomplete KD of channels versus total genetic elimination. I think this research will be of interest to those in the fields of fear, ion channels, and synaptic plasticity; however I think there needs to be revisions before acceptance.

Major Concerns:

  1. Validation of knockdown
    1. I believe confirming actual reduction of protein expression by Western blot would add an extra layer of rigor to this study, ensuring that electrophysiologic findings are not just from altered function.
  2. T-type channel investigation (voltage-clamp)
    1. The authors only reported a reduction in current amplitude during their voltage-clamp investigation of the knockdown of t-type Ca2+ channels. At the least, I think an IV plot in the supplemental material will help show that they are indeed looking at t-type Ca2+ channels. Additionally, because the authors state that t-type calcium channels, with their relatively low activation threshold, it would be interesting to know if knockdown affects input resistance at resting potentials. I did try to see if they reported this in their cited studies of t-channels (20,22) but did not find anything. As this is their first report on the effects of knockdown, more information, beyond amplitude, should be reported.
  3. Fear extinction
    1. The authors cite a paper that reported that reduced burst firing in the thalamus promotes fear extinction. It seems that this investigation was a perfect opportunity to validate or invalidate previous findings that utilized somewhat different methods of promoting and inhibiting burst firing. I would like an explanation as to why extinction was not included as part of this investigation.
  4. Abstract content
    1. The abstract isn’t that useful in understanding the overall breadth or depth of the investigation. I suggest removing lines about behavior and brain areas that do not have to do with fear or the thalamus (save this for the introduction) and use some of your allotted words to report actual data and methods in the abstract.

Minor Suggestions and Questions that should be addressed:

  1. 1 is known to play key role in auditory perception. It wasn’t clear if the odor cue was also associated with the shock in the behavioral experiments. How can the authors ensure or explain that fear conditioning is changed and not the perception of the training sound?\
  2. Experimental timing: how long after transfection were patch-clamp recordings made?
  3. There are at least two incorrect citations:
    1. Line 62, #33 does not correspond with statement
    2. Line 64, #13 perhaps should be #12?
    3. Line 67 #21, perhaps should be #22?
    4. I suggest looking at all citations in the paper, as I haven’t had time to apply this rigor to the discussion.
Comments on the Quality of English Language

Overall, the paper was well organized, however the whole narrative should be carefully reviewed to ensure it conveys their intended message. For instance, in the introduction, it says something like "outside of these regions, the role of other areas has not been investigated," but then goes on to show that the thalamus has in fact been investigated in this context. A change of wording in this section could offer clarity. Additionally there are some minor errors in the writing outlined below: 

  1. There are some confusing changes of tense in the writing:
    1. Line 32: “were not extensively studied” implies that it was not studied in the authoris investigation. I suggest change to “have not been extensively studied.”
    2. Line 256: “was not changed” likewise implies that this was reported as part of this investigation.
  2. Minor grammatical or spelling errors:
    1. Line 32-33: needs “an” to read “although an early study showed…”
    2. Line 62: “genetical” is not a conventional term in science writing
    3. Line 163 “conditioning is misspelled”
    4. Line 173 “expression” is misspelled

Author Response

This article describes an attempt by Stamenic TT et al to link behavioral changes with their previous findings that t-type Ca channels control excitability in the central thalamus. They use ex vivo electrophysiology and whole animal behavior protocols with an overall goal of an increased understanding of the mechanisms of fear conditioning and extinction in mammals. While the thalamus has been implicated in the process of fear memory consolidation, there is a paucity of investigations regarding mechanisms.

In this investigation, Stamenic TT find that knockdown of T-type calcium channels does not affect tonic firing of action potentials but does reduce burst firing and low-threshold calcium spikes. In their behavior experiments they find that both global KO and targeted KD of t-channels increase in freezing during the testing phase. There were curious differences in freeze response during the training phase that can likely be explained by incomplete KD of channels versus total genetic elimination. I think this research will be of interest to those in the fields of fear, ion channels, and synaptic plasticity; however, I think there needs to be revisions before acceptance.

We thank to the reviewer for constructive and thoughtful comments. We addressed most of the comments (reply below).

Major Concerns: 

  1. Validation of knockdown
    1. I believe confirming actual reduction of protein expression by Western blot would add an extra layer of rigor to this study, ensuring that electrophysiologic findings are not just from altered function.

Unfortunately, Western blot is not a reliable method due to inconsistent antibodies for CaV3.1 T-channels. Hence, as additional experiment for the validation of our approach we performed qPCR. The qPCR results are now presented on Figure 1 and they nicely complement our electrophysiological study.

  1. T-type channel investigation (voltage-clamp)
    1. The authors only reported a reduction in current amplitude during their voltage-clamp investigation of the knockdown of t-type Ca2+ channels. At the least, I think an IV plot in the supplemental material will help show that they are indeed looking at t-type Ca2+ channels. Additionally, because the authors state that t-type calcium channels, with their relatively low activation threshold, it would be interesting to know if knockdown affects input resistance at resting potentials. I did try to see if they reported this in their cited studies of t-channels (20,22) but did not find anything. As this is their first report on the effects of knockdown, more information, beyond amplitude, should be reported.

We agree with the reviewer and in the supplemental figure added data from our voltage-clamp experiments (IV plots). We also added resting membrane potential and input resistance data.

  1. Fear extinction
    1. The authors cite a paper that reported that reduced burst firing in the thalamus promotes fear extinction. It seems that this investigation was a perfect opportunity to validate or invalidate previous findings that utilized somewhat different methods of promoting and inhibiting burst firing. I would like an explanation as to why extinction was not included as part of this investigation. 

We agree with the reviewer that is an important issue. Unfortunately, at the time of the experiments we did not perform extinction tests but were more focused on investigation of the role of the thalamic T-channels in acquisition and expression of the fear. We added sentence about future experiments that should involve extinction phase.

  1. Abstract content
    1. The abstract isn’t that useful in understanding the overall breadth or depth of the investigation. I suggest removing lines about behavior and brain areas that do not have to do with fear or the thalamus (save this for the introduction) and use some of your allotted words to report actual data and methods in the abstract.

We agree with the reviewer, and we changed the abstract accordingly.

Minor Suggestions and Questions that should be addressed:

  1. CaV3.1 is known to play key role in auditory perception. It wasn’t clear if the odor cue was also associated with the shock in the behavioral experiments. How can the authors ensure or explain that fear conditioning is changed and not the perception of the training sound?

As per our standard protocol we always use the ethanol between animals during testing and tried to standardize odor cue so that it is always the same – during habituation, conditioning and testing. We agree with the reviewer, we cannot know if the first tone is the one to induce more freezing in CaV3.1 KO animals. Also, we observed baseline difference in freezing between WT and CaV3.1 KO animals. We discussed this more in the manuscript.

Because of the known role of the CaV3.1 T-channels in auditory perception we chose to have just 2 tone/shock pairings instead of standard protocols that are using 5 and more pairings.

  1. Experimental timing: how long after transfection were patch-clamp recordings made?

We waited 3 weeks after the surgery to perform patch-clamp experiments. We added more details about this in the text.

  1. There are at least two incorrect citations:
    1. Line 62, #33 does not correspond with statement
    2. Line 64, #13 perhaps should be #12?
    3. Line 67 #21, perhaps should be #22?
    4. I suggest looking at all citations in the paper, as I haven’t had time to apply this rigor to the discussion.

We thank the reviewer for pointing to this inconsistency, we now checked and corrected all the references.

Reviewer 2 Report

Comments and Suggestions for Authors

Stamenic et al. reported a role of CaV3.1 T-type calcium channels in contextual fear conditioning. However, the approach used and interpretation of results have major shortcomings that, in my view, precludes drawing a conclusive answers for the problems being addressed in this study. 

  1. Straight KO of the CaV3.1 gene in the whole brain prevents meaningful interpretation of the role of this particular T-channel gene in the thalamus. A targeted, region specific knockout approach makes more sense.
  2. I appreciate the use of targeted AAV-mediated knockdown by shRNA. However, the "validation" attempt will not confirm successful knockdown of this gene. There are a lot of possible off-targets from shRNA knockdown that can potentially produce the same physiological responses. RNAscope visualization of successful knockdown is a more reasonable approach here. Without a confident confirmation of CaV3.1 knockdown, it's hard to "trust" and interpret the subsequent findings due to potential secondary off-target effect of shRNA knocking down the other channels
  3. The results from behavioral paradigm are confusing. Why does the presence of tone doesn’t produce fear response, while the absence of tone still produce fear response in the KD mouse? This would suggest the phenotype is not related to fear response. Can the author explain?
  4. Again, similar to point 3 above, why the tone in Day 2 (at a different context) does not produce a freezing response in KD mouse? Supposed a specific KD at the thalamus should still elicit fear response? Can the author explain this with relevance to the role of CaV3.1 in thalamus?
  5. The paradigm used is not the widely used standard paradigm in the field. For most other studies 5 or more tones are used, is there a reason only 2 tones are used?

Author Response

Overall, the paper was well organized, however the whole narrative should be carefully reviewed to ensure it conveys their intended message. For instance, in the introduction, it says something like "outside of these regions, the role of other areas has not been investigated," but then goes on to show that the thalamus has in fact been investigated in this context. A change of wording in this section could offer clarity. Additionally, there are some minor errors in the writing outlined below: 

  1. There are some confusing changes of tense in the writing:
    1. Line 32: “were not extensively studied” implies that it was not studied in the authoris investigation. I suggest change to “have not been extensively studied.”
    2. Line 256: “was not changed” likewise implies that this was reported as part of this investigation.
  2. Minor grammatical or spelling errors:
    1. Line 32-33: needs “an” to read “although an early study showed…”
    2. Line 62: “genetical” is not a conventional term in science writing
    3. Line 163 “conditioning is misspelled”
    4. Line 173 “expression” is misspelled

We corrected all mentioned language errors, and we also used MDPI Author Services to improve our English.

Stamenic et al. reported a role of CaV3.1 T-type calcium channels in contextual fear conditioning. However, the approach used and interpretation of results have major shortcomings that, in my view, precludes drawing a conclusive answers for the problems being addressed in this study. 

  1. Straight KO of the CaV3.1 gene in the whole brain prevents meaningful interpretation of the role of this particular T-channel gene in the thalamus. A targeted, region specific knockout approach makes more sense.

We completely agree with the reviewer, and for future experiments we will use region specific KO approach instead of shRNA. However, it this case as a proof of concept we used global KO animals. We agree that this approach is not ideal, and we stated that this can be lack of the study, having in mind compensatory changes that global deletion of the channel can have.

  1. I appreciate the use of targeted AAV-mediated knockdown by shRNA. However, the "validation" attempt will not confirm successful knockdown of this gene. There are a lot of possible off-targets from shRNA knockdown that can potentially produce the same physiological responses. RNAscope visualization of successful knockdown is a more reasonable approach here. Without a confident confirmation of CaV3.1 knockdown, it's hard to "trust" and interpret the subsequent findings due to potential secondary off-target effect of shRNA knocking down the other channels.

We agree that the use of the shRNA is not ideal one, with the potential off targets but we still think that it is a valid, highly used approach. Another alternative would be use of the CaV3.1 flox animals injected with the cre-dependent virus to generate region-specific KO animals. However, we think that even with region specific KO animals we would have compensatory changes – probably more than in comparison to the shRNA approach. To functionally validate KD approach, we did voltage-clamp experiments and show that T-currents are indeed reduced. In our current-clamp experiments we observed reduction of rebound burst firing – that is highly dependent on T-channel activity after hyperpolarization of the neurons, but not tonic firing that is more dependent of other potential targets. In revision we included IV curves and PCR data as additional validation. We also included sentence about the lack of the used approach.

  1. The results from behavioral paradigm are confusing. Why does the presence of tone doesn’t produce fear response, while the absence of tone still produce fear response in the KD mouse? This would suggest the phenotype is not related to fear response. Can the author explain?

In our original manuscript it was not clear that tone does produce fear response, we just didn’t observe the differences between groups. We added Figure with the Cued fear conditioning (Day 2).

  1. Again, similar to point 3 above, why the tone in Day 2 (at a different context) does not produce a freezing response in KD mouse? Supposed a specific KD at the thalamus should still elicit fear response? Can the author explain this with relevance to the role of CaV3.1 in thalamus?

We added figure and explained the Cued Fear conditioning in both KO and KD animals.

  1. The paradigm used is not the widely used standard paradigm in the field. For most other studies 5 or more tones are used, is there a reason only 2 tones are used?

Having in mind the role of T-channels in auditory perception (Bayazitov IT, Westmoreland JJ, Zakharenko SS. Forward suppression in the auditory cortex is caused by the Ca(v)3.1 calcium channel-mediated switch from bursting to tonic firing at thalamocortical projections. J Neurosci. 2013 Nov 27;33(48):18940-50. doi: 10.1523/JNEUROSCI.3335-13.2013. PMID: 24285899; PMCID: PMC3841456.), we used the altered protocol with just two shocks instead of more, so our conditioning to the tone was not that strong in WT or control (scrambled shRNA) mice. In day two, tone as a factor was statistically significant in both experiments but with no differences between tested groups (we added Figure with the cued conditioning). Since we did not use standard protocol (with more tone pairings) in our experiments, conditioning to the context was more evident and difference between the groups were observed.

Round 2

Reviewer 2 Report

Comments and Suggestions for Authors

I appreciate the authors address the concerns of potential non-specific knockdown. I recommend the authors to discuss the difference in behavioral paradigms used compared to standard paradigms (i.e. >5 tones vs 2 tones) in the Discussion section. This will improve the manuscript by allowing audience to understand the rationale of using a "special" behavioral paradigm. This will not require additional experiments but merely an inclusion of several sentences in the Discussion section.

Author Response

We thank to the reviewer for the comment. We added a few sentences in the Discussion section regarding the difference in behavioral paradigms we used compared to standard ones.